# Automated Interlayer Wall Height Compensation for Wire Based Directed Energy Deposition Additive Manufacturing

**DOI:** 10.3390/s23208498

**Published:** 2023-10-16

**Authors:** Jian Qin, Javier Vives, Parthiban Raja, Shakirudeen Lasisi, Chong Wang, Thomas Charrett, Jialuo Ding, Stewart Williams, Jonathan Mark Hallam, Ralph Tatam

**Affiliations:** 1Welding and Additive Manufacturing Centre, Cranfield University, Cranfield MK43 0AL, UK; chong.wang1@cranfield.ac.uk (C.W.); jialuo.ding@cranfield.ac.uk (J.D.); s.williams@cranfield.ac.uk (S.W.); 2WAAM3D Limited, Thornton Chase, Linford Wood, Milton Keynes MK14 6FD, UKp.raja@waam3d.com (P.R.); s.lasisi@waam3d.com (S.L.); 3Centre for Engineering Photonics, Cranfield University, Cranfield MK43 0AL, UK; t.charrett@cranfield.ac.uk (T.C.); jonathan.hallam@cranfield.ac.uk (J.M.H.); r.p.tatam@cranfield.ac.uk (R.T.)

**Keywords:** wire-based directed energy deposition additive manufacturing (w-DEDAM), part quality monitoring and control, interlayer wall height compensation

## Abstract

Part quality monitoring and control in wire-based directed energy deposition additive manufacturing (w-DEDAM) processes has been garnering continuous interest from both the academic and industrial sectors. However, maintaining a consistent layer height and ensuring that the wall height aligns closely with the design, as depicted in computer-aided design (CAD) models, pose significant challenges. These challenges arise due to the uncertainties associated with the manufacturing process and the working environment, particularly with extended processing times. To achieve these goals in an industrial scenario, the deposition geometry must be measured with precision and efficiency throughout the part-building process. Moreover, it is essential to comprehend the changes in the interlayer deposition height based on various process parameters. This paper first examines the behaviour of interlayer deposition height when process parameters change within different wall regions, with a particular focus on the transition areas. In addition, this paper explores the potential of geometry monitoring information in implementing interlayer wall height compensation during w-DEDAM part-building. The in-process layer height was monitored using a coherent range-resolved interferometry (RRI) sensor, and the accuracy and efficiency of this measurement were carefully studied. Leveraging this information and understanding of deposition geometry, the control points of the process parameters were identified. Subsequently, appropriate and varied process parameters were applied to each wall region to gradually compensate for wall height. The wall height discrepancies were generally compensated for in two to three layers.

## 1. Introduction

Wire based directed energy deposition additive manufacturing (w-DEDAM) is one of the most popular additive manufacturing technologies, which uses electric arc energy, electron beam energy, or laser energy to melt the metal material or alloy to deposit large, fully dense, three-dimensional (3D), near-net-shape metallic components [1,2]. For the successful operation of this technology, it is essential to continually monitor and control a set of parameters and requirements throughout the construction period. This ensures that the final product not only meets but also maintains the minimum necessary quality standards. Efficient quality monitoring and control are essential to achieve this target. Various sensor technologies have been utilised to address the challenges related to monitoring the DEDAM process and ensuring part quality. These technologies include vision, laser, acoustic, and thermal sensing [3]. 

For example, Vandone et al. [4] adopted a passive vision system in which a complementary metal–oxide–semiconductor (CMOS) camera and image processing software were used to monitor and control the geometry of the deposition. This camera system was also used to monitor the melt pool of the cladding process [5]. Taking advantage of the image processing, we extracted the width of the melt pool for monitoring purposes. Similarly, Zhao et al. [6] also implemented a vision system that was able to measure the melt pool geometry. This becomes even more effective when it is combined with a spectrum detector, enabling the capture of material changes such as the addition of hydrogen and foreign substances during the deposition process. Another quality monitoring methodology was implemented by Xiong et al. [7] by using the computer version (CV) methodology. This study presents the design of a novel multichannel monocular vision sensor to monitor different geometric sizes. The process camera is based on Marr–Hildreth algorithms [8], based on the method of detecting edges in digital images. A boundary extraction algorithm is used to determine the boundaries between the molten pool and the previous layer. Height calibration is performed with a homography matrix, and the detected heights are introduced to assist in the calibration of the layer widths. However, in studies in which CV technologies are employed as primary monitoring approaches, the measurements can be influenced by various environmental factors. These factors include lighting conditions, optical properties of the deposited materials, surface conditions, and glare from the melt pool. 

To provide an accurate and real-time deposition geometry under the crucial w-DEDAM working environment, a DEDAM-specific sensor needs to be designed and developed. One of the main limitations for the part quality monitoring and control study, especially the geometric control, appears when working in extreme environments becomes a critical research and industrial challenge. Many researchers have thus focused on using different sensors to obtain the geometry profile in AM processes. In the study by Tang et al. [9], a laser-based areal topography measurement sensor was used to measure the geometric signatures of the formed layer. A topography control strategy was performed by processing and analysing the 3D surface profile data. Abe et al. [10] adopted thermal sensing to monitor and control the deposition, and a heat-input condition control system was proposed and developed. This system controlled the voltage, a heat-input condition, which was applied from the power source according to the temperature of the building structure. The temperature of this structure was measured using a radiation thermometer during the deposition process. To determine the optimal voltage value, the temperature of the relationship between the build structure and the deposited metal shape was estimated by numerical simulation [11]. 

When the geometry profile can be measured with precision, it can be controlled using various methods to ensure that a stable and accurate profile is maintained. According to research conducted by Binega et al. [12], the correct geometric controls are based on examination by comparing the profiles of single- and multilayer objects estimated during the DEDAM process with the reference profiles obtained by laser line scanning. Their proposed methodology comprises a real-time scanning of each deposited track’s profile, followed by an online extraction of the track’s geometry and, finally, an online plotting and comparison of the as-designed and as-built models. In this methodology, data analysis following real-time scanning, such as geometry discrepancy estimation and online plotting of the as-built model, is attained for both single- and multilayer objects. Furthermore, a qualified AM part also depends on precise and efficient process control. Ščetinec et al. [13] used a different control method, which combined a proportional–integral–derivative (PID) controller to control the wire feed speed and vision system. An online layer height control and in-process toolpath replanning system was developed for the gas metal arc (GMA)-based wire arc additive manufacturing (WAAM) system. WAAM is one group of w-DEDAM technologies, in which the arc is the main power source to melt the wire and deposit the part. This research enables better geometric accuracy when depositing large parts of the shell. Compared to PID-based control, Mu et al. [14] developed a model predictive control (MPC) using an autoregressive exogenous (ARX) model. Two control strategies were developed and evaluated: a PID algorithm, and a multi-input multi-output model predictive control (MPC) algorithm. After each layer of deposition, the deposited geometry was measured using a laser scanner. These measurements were compared against the CAD model, and then the controller compensated for geometric errors, generating a new set of welding parameters for the next layer. The MPC algorithm, combined with a linear autoregressive modelling process, updated the welding parameters between successive layers by minimising a cost function based on sequences of input variables and predicted responses [15]. 

Existing research on monitoring and controlling the geometry profile has not considered the responses of the process when there are changes in the input parameters during deposition. However, the w-DEDAM process is highly sensitive to any alterations in the process parameters, necessitating a comprehensive understanding of the process. This paper aims to investigate the behaviour of geometry during wall height control, and to develop an effective control strategy for wall height compensation. To achieve this, a cutting-edge process geometry measurement sensor called the coherent range-resolved interferometry (RRI) sensor is utilised for measuring the wall height [16]. This study encompasses an analysis of the process, followed by the presentation of the control strategy and the corresponding results. The structure of this paper is as follows: In Section 2, the experimental setup is introduced, along with an explanation of the principles behind the RRI sensor. Section 3 presents the observations from the process study, including the transition study. The wall height compensation control strategy and its performance are presented in Section 4. Finally, Section 6 concludes the investigation, summarising the findings and insights gained from the discussion presented in Section 5.

## 2. System Setup and RRI Sensor

### 2.1. System Description

The w-DEDAM system utilised in this research is a robotic plasma-based WAAM system. The system, depicted in Figure 1 (left), comprises a KUKA industry six-axis robot, an end effector that integrates a torch and sensors, a Dinse push-and-pull wire feeder, and an EWM 350 plasma power source. The end effector, also shown in Figure 1 (right), consists of a plasma torch, a local shielding mechanism (so-called WAAM shield) that facilitates the flow of the shielding gas through laminar flow baffles, a wire feeder nozzle, a process camera, and an RRI sensor used to capture layer height data. The wire feeder nozzle is positioned in front of the welding torch, following the direction of deposition. Vertical control of the position of the wire feeder nozzle is achieved using a motor drive. The deposition experiments were conducted on a Ti-6Al-4V substrate measuring 500 mm × 80 mm × 6.5 mm, with a Ti-6Al-4V wire of 1.2 mm diameter.

The RRI sensor was in line with the welding direction with an offset of 25 mm behind the tip of the welding torch. Figure 2 shows the schematic of layer height measurement ($H_{wall}$) using the RRI sensor. 

The RRI measured the distance ($M_{rri}$) between the fibre tip (inside the collimator) and the target surface. Working with the position of the robot ($P_{robot}^{z}$), the wall height and layer height were calculated as shown below. More technical details about the RRI sensor are introduced in the next subsection.
$$H_{wall}=P_{robot}^{z}-H_{rri}\;$$

### 2.2. Coherent Range-Resolved Interferometry (RRI) Sensor

As discussed in the previous subsection, the RRI sensor was pivotal in measuring the layer height for this research. It is typically challenging to obtain in-process measurements of layer heights in a notably bright environment, such as those involving arc- or laser-power-based processes, using conventional cost-effective distance sensors such as triangulation or confocal sensors [17]. Furthermore, both triangulation and confocal sensors require relatively large sensor heads, which complicates their placement close to the weld pool. This is the rationale behind the proposal of an RRI sensor [18] as a technique that allows for in-process layer height measurements in this research. In this technology, as shown in Figure 3, a diode laser’s optical frequency is sinusoidally modulated, and the resulting light is both supplied to and collected from the target using optical fibres. The light reflected off the target (usually the layer surface) interferes with reference light reflected from the fibre tip. The resulting light is demodulated for a range of fibre-tip-to-target-surface distances covering approximately 100 mm using pre-computed time-variant carrier signals that correspond to the expected interference signals. The absolute distance of the target is then determined by evaluating the return signal intensity as a function of range using a Gaussian peak fit. The complete mathematical derivation is presented in Ref. [18].

The proposed RRI sensor offers a robust and versatile method for distance measurements. Implemented in a reduced-size configuration for a more compact assembly head, it is capable of achieving a measurement resolution of better than 100 μm and a working range exceeding 100 mm. As an added benefit, given its coherent nature, it is inherently insensitive to arc light, allowing for measurements close to the weld pool [19]. Furthermore, to obtain accurate measurements, the RRI sensor can also be calibrated using a calibration block—a predesigned precision block with stairs of varying known dimensions. The collimator is placed at a working distance from the surface of each stair landing, and the range measurement is taken. The ratio of the measured range against the known stair dimension is used to calibrate the sensor.

By modulating the laser injection current at a frequency of 410 kHz, a 5 mW, 1550 nm diode laser undergoes sinusoidal wavelength modulation. The laser light is channelled through a fibre-optic circulator and a single-mode optical fibre leading to a fibre-coupled adjusted 10 mm diameter collimator lens, which delivers a focused beam to the target, as illustrated in Figure 3. The light reflected by the target merges with (because it is coherent with) the Fresnel reflection at the fibre tip within the sensor head, making a length measurement from the fibre tip to the target and rendering the system entirely insensitive to down-leading fluctuations in the fibre [16,18]. 

Field-programmable gate array (FPGA)-based signal processing hardware demodulates the interference signals resulting from the return light guided to a photodetector. To obtain distance measurements with an RRI sensor [19], the strength of the return signal is evaluated as a function of range, and the peak of the return signal is fitted using a Gaussian function. Averaged output data are gathered at 3.2 kHz. The sensor head is directly mounted on the end effector. To ensure that the layer height measurement is conducted outside the weld pool, the laser spot is positioned 25 mm behind the centre of the weld pool, as detailed in the previous section. The RRI sensor is employed to measure the distance from the fibre mounted on the end effector to the top of the wall. An initial measurement of the distance to the substrate is made, and successive measurements at each layer allow the increase in the height of the wall relative to the substrate to be determined. This representation is used to select control points (CPs). Data collected by the RRI sensor revealed that the start and end points did not share the same height, indicating a slight inclination. Consequently, calibration was performed by adjusting each data point according to the degree of inclination detected in the data [20].

## 3. Process Study on In-Layer Wall Height Variation and Control

This section provides a detailed explanation of the experimental study on the variation and control of in-layer wall height. The plan for the experiment is initially presented, followed by the observation of the experiment. Furthermore, this section presents an additional investigation into the transit area when different process parameters are applied during the deposition process. 

### 3.1. Experimental Plan

Before the experimental plan, the range of plasma process parameter selection needs to be clarified. The process parameters used in this experiment were recommended in the study by Martina et al. [20]. This process study was carried out using a comprehensive set of experiments with the material Ti-6Al-4V. It also investigated the geometry of various basic plasma-based WAAM process parameters, in terms of wire feed speed, current, and travel speed. Since the proposed research focuses on layer height control, the parameters chosen in this research require a reduced wall-width effect but a significant wall-height effect. The process parameter range was selected according to Table 1, in which the wall width has less than 10% differences according to Martina et al.’s [20] work. For a travel speed (TS) of 6  mm/s operating at a 200  Amp current, the effective wall width was generally constant at 9.5  mm when WFS was changed from 60  mm/s to 90  mm/s. A similar constant effective wall width was observed for currents ranging from 200  Amp to 250  Amp.

Based on the above process parameter range, single-pass walls were built with different parameter combinations, which changed the wall height in different regions on the wall. Each wall was built with constant process parameters for the first nine layers to generate the base layers. From the 10th layer to the 13th layer (experimental layers), the parameters were changed within four different regions, which are called control units (CUs). The design of the single-pass wall is shown in Figure 4. 

The single-pass wall measured a total length of 400 mm, with two control points designated to discern the regions corresponding to varying process parameters for each CU. The respective control points are specified in Table 2. The CUs used in this research were designed to be 10  mm, 20  mm, 30  mm, and 40  mm in length. These control points provide crucial indexing information for robot programming, typically employed to develop the robot program for deposition. During the layering experiments, the parameters within the CUs were distinctly set in experimental layers, with the base layer process parameters used in the remaining areas.

The process parameter setup is shown in Table 3. Based on this experimental plan, 8 single-pass walls were built to create different wall height variances in four CUs. The experiment was subdivided into 2 different sections to define the CUs across the 400  mm wall. In total, eight distinct walls were constructed to assess the variation in layer height, using the parameters outlined in Table 3. According to the process parameters previously shown in Ref. [20], due to the different wire feed speeds, the heights of the CUs in wall indices 1 to 4 were increased, while the height of the CUs in wall indices 5 to 8 were decreased. 

The initial sections escalated the layer height by boosting the WFS from 60  mm/s to 75  mm/s, and then to 90  mm/s, paired with currents of 200  Amp and 250  Amp, respectively. The subsequent section reduced the wall height by decreasing the WFS from 90  mm/s to 75  mm/s and, finally, to 60  mm/s, with currents of 200  Amp and 250  Amp. Throughout the study, a consistent TS of 6  mm/s was applied. Each of the CU experiments was conducted to assess the response to the parameter changes and assess the proficiency of the control process.

### 3.2. Experimental Observation

#### 3.2.1. Wall Height Variance in Different Control Units

The entire layer height information was captured from 8 wall depositions. The different CU parameters were applied from the 10th layer to the 13th layer. Figure 5 shows the photo of a deposition wall (wall index 1) with 13 layers, compared to the RRI sensor measurement.

Based on Figure 5, the RRI sensor measurements reveal an irregular wall of 400 mm. It is noticeable that each corresponding CU (10 mm, 20 mm, 30 mm, and 40 mm) generated by different process parameters differs from the rest of the wall. The experiment was divided into two parts. The initial part involved increasing the WFS from 60 mm/s to 75 mm/s, followed by an increase from 60 mm/s to 90 mm/s. The results are presented in Figure 6, where the CU parameters were applied from the 10th layer to the 13th layer. Upon examining the variation in layer height compared to the base layer height for each layer, it was observed that CU lengths of 40 mm, 30 mm, and 20 mm exhibited a very similar height variation. However, there was a significant difference in height variation for CU lengths of 10 mm. This can be attributed to the short transaction area caused by machine delay time and process transition time. In the second part of the experiment, the WFS was decreased from 75 mm/s to 60 mm/s and from 90 mm/s to 75 mm/s. The findings are illustrated in Figure 7. Similar to the previous observation, CU lengths of 40 mm, 30 mm, and 20 mm showed relatively closer heights compared to CU lengths of 10 mm, particularly between layers 11 and 13. It should be noted that the variation in wall height represents the average of the RRI sensor measurements for each respective region. In general, when both the current and travel speed are held constant, an increase in wire feed speed (WFS) leads to a corresponding increase in wall height, while a decrease in WFS results in a lower wall height. It is worth noting that, based on the work of Martina et al. [20], the wall width remains largely consistent when using the process parameters under consideration.

Furthermore, the transition area adjacent to each CU is clearly visible, as emphasised in Figure 8. This transition area is consistently present in all CUs and serves as a distinct shape feature when there is a change in the process parameters, particularly in relation to the wall height. Whenever adjustments are required to achieve the desired wall height, the process parameters are modified, resulting in the emergence of the transition area. A thorough investigation of the transition area is crucial to obtain precise details regarding wall height compensation, enabling accurate control over the wall height. This important study will be discussed in the next subsection. 

#### 3.2.2. Transition Area Study

Based on observations, the main factor contributing to the presence of the transition area is the response time of the process, which can be influenced by two factors: machine delay time, and process transition time [21]. The delay time of the wire feeder was measured, and various parameters (including WFS, current, voltage, and others) were recorded during the deposition process. To assess the response delay, the WFS was alternated between 4.5 mm/s and 3.6 mm/s. Figure 9 illustrates that the response delay of the WFS is typically less than 0.5 s, which generates less impact for the change in geometry. 

To gain a comprehensive understanding of wall height control, several aspects, such as the impact of the transition time on parameter changes, needed to be investigated. In this study, additional experiments were designed and conducted, and the experimental plan is presented in Table 4. 

The methodology for assessing the impact of process parameters on transition time followed the same approach as our previous experiments. Various parameters were applied to the control units (CUs), and the dimensions of the transition area were then measured. Based on the robot’s travel speed, we were able to calculate the transition time. According to Table 4, there are three aspects that are considered during the transition time: travel speed, current, and the ratio of WFS to TS [2]. Two different travel speed values were used to examine the effect of TS: 8 mm/s and 10 mm/s. As shown in Figure 10a, it was observed that as the travel speed increased, the transition time decreased. This trend was more pronounced for a travel speed of 10 mm/s compared to 8 mm/s. A similar study was conducted to analyse the effects of different currents and the ratio of WFS to TS. However, when the effect of current was analysed, no predictable correlation was observed, as shown in Figure 10b. Interestingly, no correlation was observed between different lengths of CUs, indicating that there was no geometric relationship between the size of the CU and the changes in the parameters. Figure 10d illustrates the relationship between CU length and transition time. Furthermore, it was observed that for different changes in the WFS-to-travel-speed ratio, the transition time (Figure 10c) ranged from 2 to 3 s. When the ratio of WFS to TS increased, either due to higher WFS or lower TS, the transition time also correspondingly increased. As shown in Figure 10c, when the WFS-to-TS ratio was 2.0, the recorded transition time deviated from the trend line established by the other four recorded results. This anomaly can be attributed to experimental uncertainties, such as measurement errors, time-recording discrepancies, and inconsistencies in monitoring devices. Further discussion of these experimental uncertainties can be found in Section 5.

The transition areas observed are also influenced by the shape of the plateau. This shape is particularly evident for CU lengths of 30 mm and 40 mm at travel speeds ranging from 6 mm/s to 10 mm/s, as indicated in Table 5. For a CU length of 20 mm, at a travel speed of 6 mm/s, it appears as a peak. However, with a higher travel speed, the complete transition is clearly visible, presenting a plateau shape. In the case of a CU length of 10 mm, the length is too short for the transition to occur fully, which is why it appears as a peak. Similar effects were observed for travel speeds of 8 mm/s and 10 mm/s, as well as for different currents. 

## 4. Interlayer Geometry Compensation

### 4.1. Initial Wall Compensation Validation

Before introducing the complete method for controlling the wall height, the idea of modifying the parameters on the initial CPs (control points) to restore the desired wall height needed to be verified. In this section, the previous walls were continuously constructed by exchanging the CU process parameters and the base layer process parameters, as specified in Table 5. Starting from the 14th layer, the base parameters used were WFS: 75 mm/s, TS: 6 mm/s, and current: 250 Amp, while the CU parameters used were WFS: 90 mm/s, TS: 6 mm/s, and current: 250 Amp. 

Table 6 illustrates that the variations in wall shape and height consistently decreased after applying the compensation parameters. Prior to the correction, the average variation in the wall height, defined by the average difference between the maximum and minimum wall height, was −0.64 mm. After three consecutive layer corrections, the standard variation was reduced to 0.06 mm at layer 15. 

Figure 11 shows that the CUs with lengths of 10  mm and 20  mm achieved an absolute average variation of less than 0.25  mm, exhibiting a faster response compared to the CUs with lengths of 30  mm and 40  mm. In the second correction layer (layer 15), all CUs achieved an absolute average variation below 0.5  mm. However, further compensation beyond this layer resulted in a higher wall height (i.e., overcompensation) at the CUs. 

### 4.2. Closed-Loop Wall Height Compensation Control Flow

Based on the findings of this study, we propose a closed-loop layer-by-layer process for wall height compensation, as depicted in Figure 12. Upon the deposition of the current layer, the geometry profile is captured by the RRI sensor, which accurately determines the current layer’s profile and wall height with the necessary resolution. These data are then compared with the desired wall height derived from the original path planning, allowing the calculation of the deviation for the subsequent layer. Taking into account this deviation and the original path planning, an optimised height for the next layer is derived, subsequently forming the basis for a revised path planning strategy. This strategy is further reinforced by the fundamental process parameter settings. Our control method effectively bridges the design and production phases, facilitating the generation of a robust production plan for the subsequent layer.

Table 7 below presents the CPs ascertained by the RRI sensor. As previously discussed in Table 3, our initial plan proposed 10 CPs spanning the 400  mm wall. Each of these CPs was pinpointed by our algorithm, informed by the discrepancies detected by the RRI sensor. Due to the deliberate variation in wall height for the 14th layer, these CPs were distinctly delineated, as evidenced in Table 4 (which outlines the initial CPs). A comparison between these initial CPs and those identified later allowed for a thorough validation of the algorithm’s precision in discerning CPs. On average, the errors hovered around 1.7  mm. An intriguing observation was that with each subsequent layer, the distance between the CPs expanded, while the number of control points diminished. This consistent pattern underscores the methodical approach ingrained in the CP generation process.

### 4.3. Wall Height Compensation Results

The impact of the control mechanism, as outlined in Table 7, became evident in the layer formations, which are illustrated in Table 8. The average fluctuation in wall height was consistently reduced with each successive controlled layer. Specifically, the mean deviation in wall height declined from 1 mm at the 13th layer to less than 0.25  mm by the conclusion of the 15th layer (also known as the 2nd control layer). This trend was found to persist in the layers that followed. The control process demonstrated stability, successfully maintaining the average wall height variation within a tight limit of 0.25  mm for all subsequent layer depositions. 

Furthermore, it is worth noting that control unit (CU) lengths of 10  mm and 20  mm achieved an average variation nearing zero. However, a closer examination of the initial CU regions, as shown in Figure 13, reveals that a CU length of 10 mm required more layers to achieve stability. In contrast, the initial CU segments of 20  mm, 30  mm, and 40  mm demonstrated control by the second control layer. 

The data gathered by the RRI sensor throughout these experiments demonstrated reliability and repeatability, thereby establishing it as one of the most suitable methods for measuring layer heights. This assertion is supported by the experimental results visualised in Figure 14. As the quantity of control layers increased, the variation in wall height consistently maintained a limit of 0.25 mm. 

For single-pass multilayer walls, the methodology of this study proved effective in controlling the wall height. However, it is worth noting that the wall height control in control unit (CU) regions of 10 mm and 20 mm was less effective compared to larger CU sizes. This was due to insufficient time for the wall height transition to occur, which practically implies that the travel speed of the welding process outpaced the time necessary for the height change to mirror the selected parameter switch. To remedy this issue, more refined parameters could be incorporated into the process. The RRI sensor efficiently captured the required wall height variation. This research demonstrates the feasibility and effectiveness of layer height control using data collected by the RRI sensor, coupled with a machine learning algorithm for layer height compensation.

## 5. Discussion

The wall height control for the CU regions of 10 mm and 20 mm is comparatively less effective than for the other CU sizes. This is due to there being insufficient time for the wall height transition to occur. In practical terms, this means that the travel speed of the deposition process is faster than the time taken for the height change to reflect the chosen parameter change. However, more precise process parameters can help to solve this problem. Therefore, the method used in this study for wall height control works effectively for single-pass multilayer walls. 

The RRI sensor was effective in measuring the variation in wall height according to the requirements. This study extended this application, and we found that layer height control using the data gathered by the RRI sensor is possible and effective. Based on the experiments, the collected RRI sensor was reliable, repeatable, and proved to be a suitable method to measure layer height. There was no detectable interference with the arc, and it worked well with smooth and shiny deposited surfaces. As explained above, the average wall height deviation achieved was less than 0.25 mm after the second control layer. The variation in wall height continued to be within 0.25 mm as the number of control layers increased.

The proposed approach exhibits versatility, as it can be tailored to various w-DEDAM processes. These include laser wire additive manufacturing and cold wire metal inert gas (CW-MIG) [22]. Furthermore, this methodology is adaptable to different wall deposition strategies, such as parallel and oscillation walls. Figure 15 shows a defective parallel wall constructed using the CW-MIG process. Measurements of this wall were obtained via the RRI sensor, and the CPs were identified using the proposed technique. This underlines the effectiveness and applicability of the proposed method across various AM processes and strategies. 

The compensation strategy proposed for wall height discrepancies was put into action on a defective wall over the course of the next two layers. The rectified wall is depicted in Figure 16, where the heights of the two adjusted layers are also distinctly displayed. The results from the compensated wall serve as a testament to the validity of the proposed method. The technique was used efficiently across a range of AM processes and accommodated a variety of wall shapes, further strengthening its applicability and effectiveness.

It is important to acknowledge the limitations of this research. The accuracy of our experimental results was affected by various uncertainties, including device calibration errors, measurement inaccuracies, and data recording issues. While repeated experiments could mitigate these uncertainties, the w-DEDAM process is costly, particularly when using high-value materials like titanium alloys. Therefore, the financial implications of conducting such experiments are also a topic worthy of further investigation. There are several avenues for future research. The data acquisition system for wire feed speed in the transition area could be enhanced. Moreover, there are numerous valuable research opportunities to explore for reducing the transition area with a higher-frequency data acquisition system, thereby improving the accuracy and efficiency of wall height compensation. These could include more extensive parametric studies involving different materials, as well as the exploration of alternative part-building strategies, such as oscillation and parallel approaches.

## 6. Conclusions

In conclusion, this research primarily underscores the utility of the innovative in-process RRI sensor for the w-DEDAM process, particularly for capturing wall height information. The study carefully examined and developed an understanding of in-layer wall height variations and the method for wall height control. The key findings of this research can be summarised as follows:The RRI sensor has proven to be effective in capturing and measuring layer and wall shape information for w-DEDAM processes. It has the potential to handle more complex shapes of deposition walls.The layer height transition is gradual and eventually achieves a steady state after a certain transition period. This transition time is greatly influenced by the travel speed. However, elements such as the control unit length and the current or the wire feed rate/travel speed (WFT/TS) ratio were deemed to be inconsequential.Shorter defects (i.e., those below 20 mm long) were more difficult to control, but the parameter changes did not result in a stable layer height within the control unit length. Consequently, this necessitates the selection of more optimal parameters that are in tune with the height variation, via a mathematical model.

## Figures and Tables

**Figure 1 sensors-23-08498-f001:**
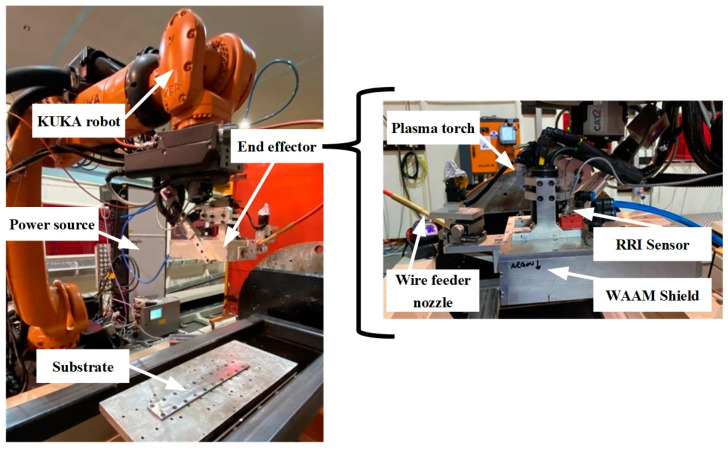
Schematic representation of the WAAM setup; (**left**) complete system, (**right**) end effector.

**Figure 2 sensors-23-08498-f002:**
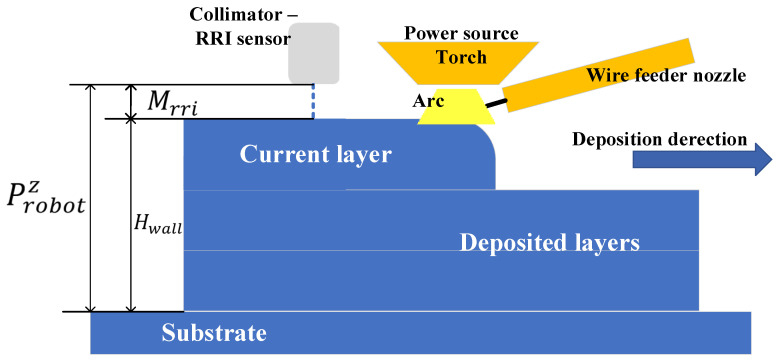
Schematic representation of the position of the range-resolved interferometry sensor, and measured distances.

**Figure 3 sensors-23-08498-f003:**
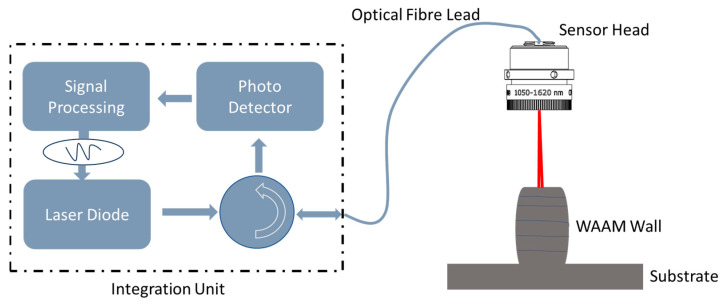
Range-resolved interferometry system.

**Figure 4 sensors-23-08498-f004:**
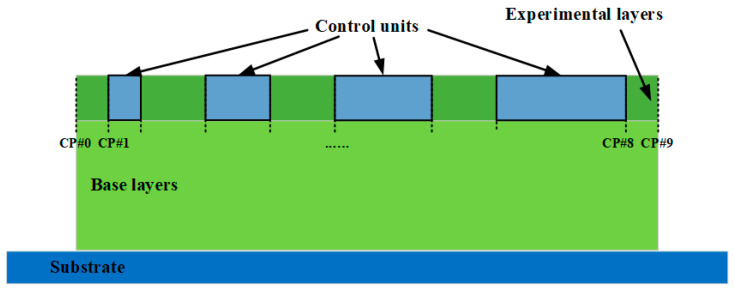
Design of the single-pass wall deposition.

**Figure 5 sensors-23-08498-f005:**
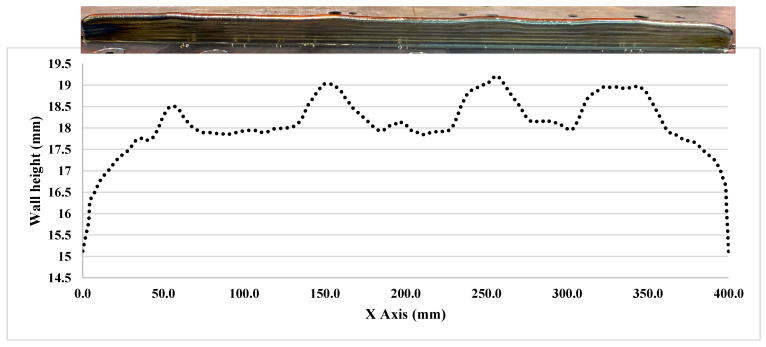
Deposition wall and range-resolved interferometry sensor measurement of wall index 1.

**Figure 6 sensors-23-08498-f006:**
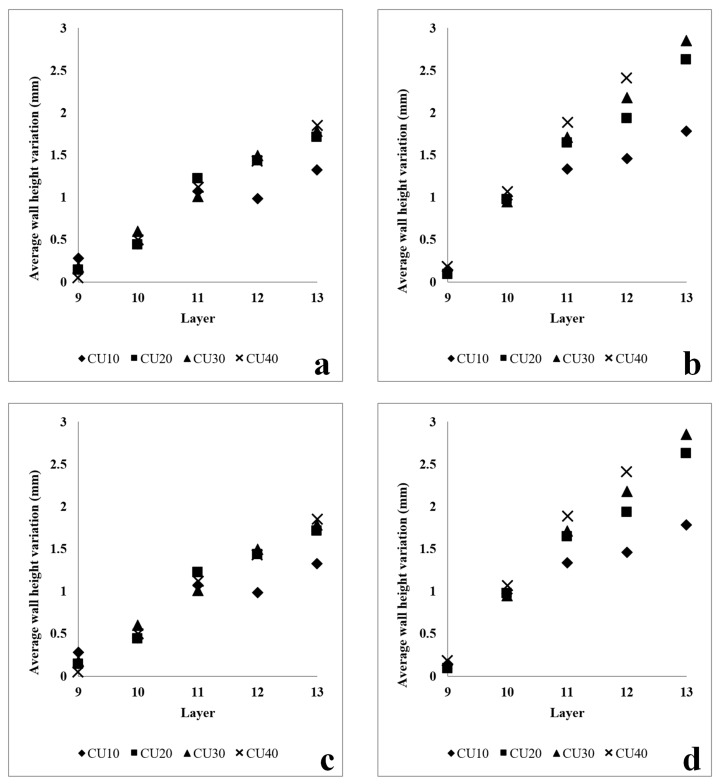
Variation in wall height with increasing wire feed speed with different process parameters. (**a**) Wall #1: Current: 200 Amp; TS: 6 mm/s. Base layer parameter: WFS: 60 mm/s. CUs parameters: WFS 75 mm/s. (**b**) Wall #2: Current: 200 Amp; TS: 6 mm/s. Base layer parameter: WFS: 60 mm/s. CUs parameters: WFS 90 mm/s. (**c**) Wall #3: Current: 250 Amp; TS: 6 mm/s. Base layer parameter: WFS: 60 mm/s. CUs parameters: WFS 75 mm/s. (**d**) Wall #4: Current: 250 Amp; TS: 6 mm/s. Base layer parameter: WFS: 60 mm/s. CUs parameters: WFS 90 mm/s.

**Figure 7 sensors-23-08498-f007:**
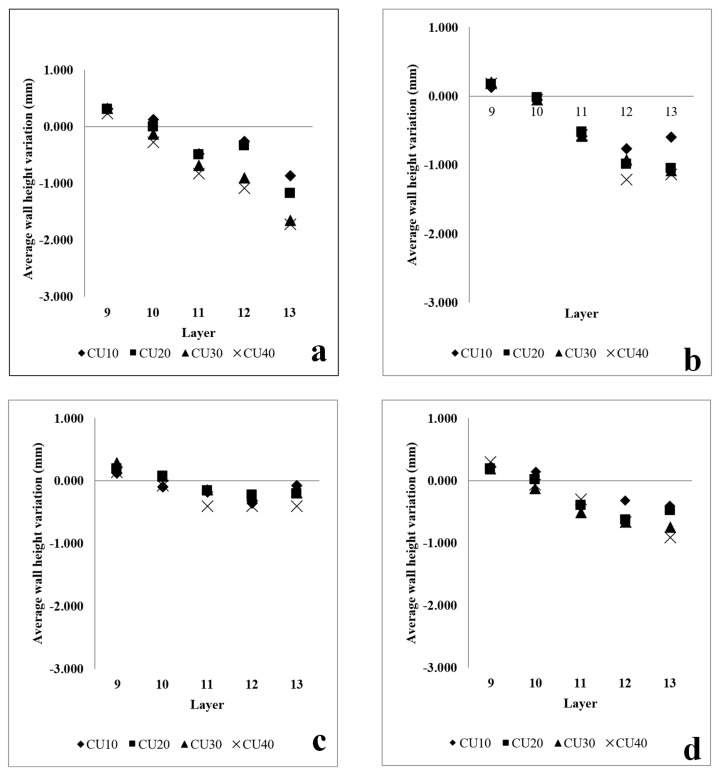
Variation in wall height with increasing wire feed speed with different process parameters. (**a**) Wall #5: Current: 200 Amp; TS: 6 mm/s. Base layer parameter: WFS: 90 mm/s. CUs parameters: WFS 60 mm/s. (**b**) Wall #6: Current: 200 Amp; TS: 6 mm/s. Base layer parameter: WFS: 90 mm/s. CUs parameters: WFS 75 mm/s. (**c**) Wall #7: Current: 250 Amp; TS: 6 mm/s. Base layer parameter: WFS: 90 mm/s. CUs parameters: WFS 60 mm/s. (**d**) Wall #8: Current: 250 Amp; TS: 6 mm/s. Base layer parameter: WFS: 90 mm/s. CUs parameters: WFS 75 mm/s. .

**Figure 8 sensors-23-08498-f008:**
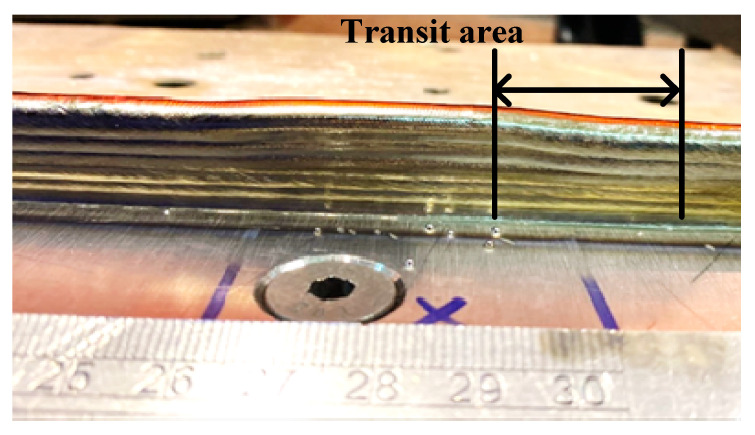
An example of a transition area.

**Figure 9 sensors-23-08498-f009:**
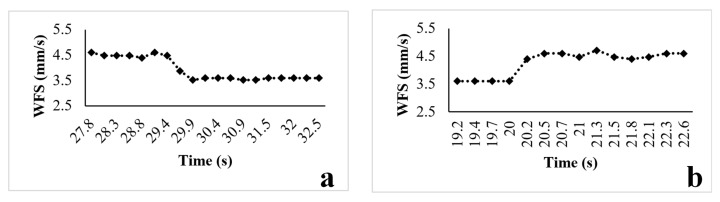
(**a**) Delay in decreasing the wire feed speed. (**b**) Delay in increasing the wire feed speed.

**Figure 10 sensors-23-08498-f010:**
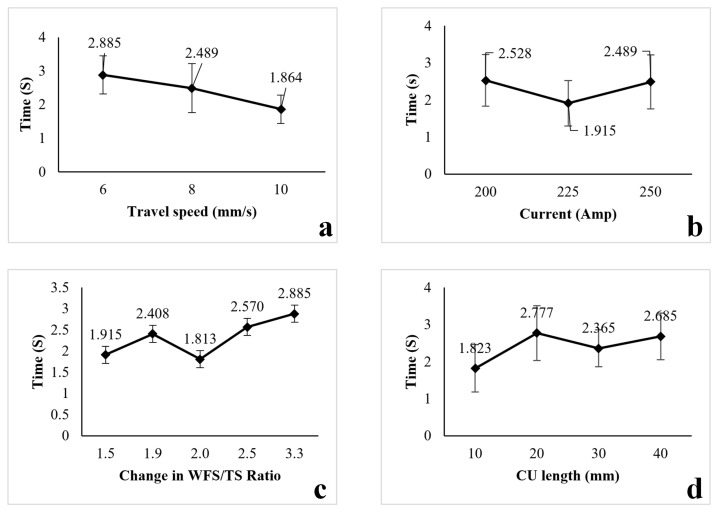
(**a**) Effect of travel speed on transition time. (**b**) Effect of current on transition time. (**c**) Effect of change in the ratio of wire feed speed to travel speed on transition time. (**d**) Effect of control unit length on transition time.

**Figure 11 sensors-23-08498-f011:**
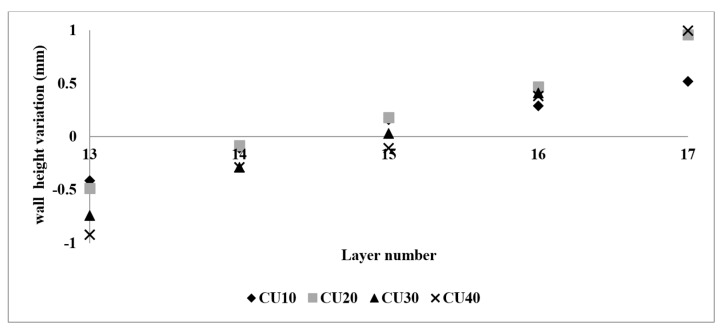
Control unit height variation of the correction test.

**Figure 12 sensors-23-08498-f012:**
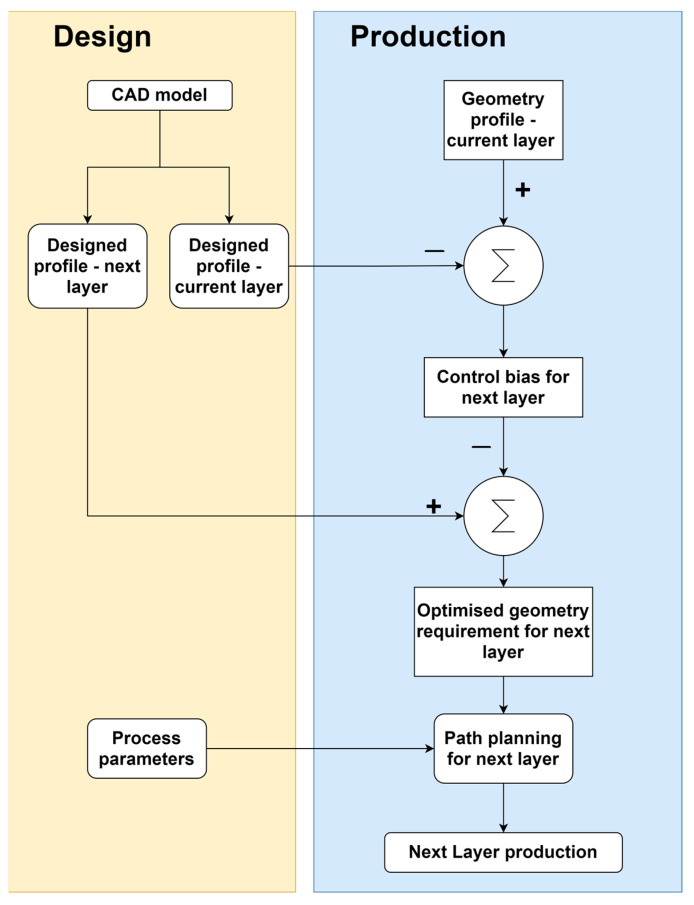
Wall height control process flow.

**Figure 13 sensors-23-08498-f013:**
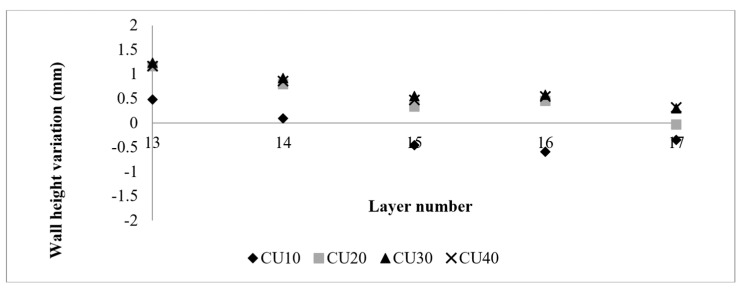
CU height variation of the correction test.

**Figure 14 sensors-23-08498-f014:**
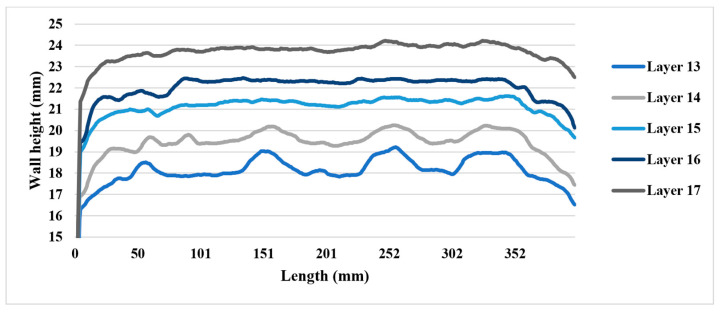
Average wall height variation of layers 13 to 17.

**Figure 15 sensors-23-08498-f015:**
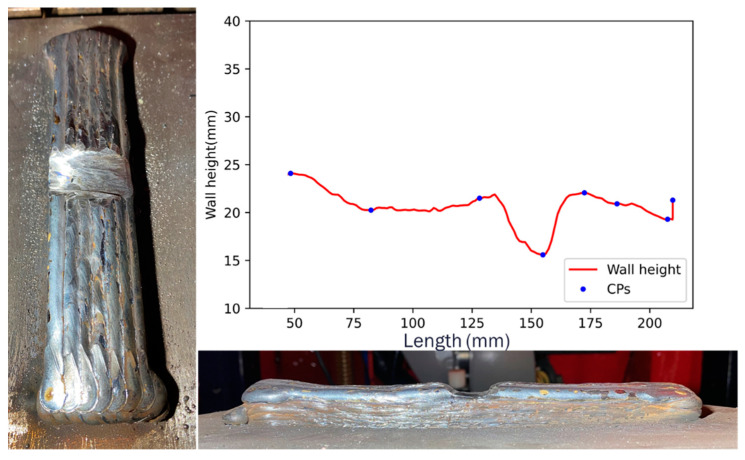
Defective parallel wall built by the CW-MIG process.

**Figure 16 sensors-23-08498-f016:**
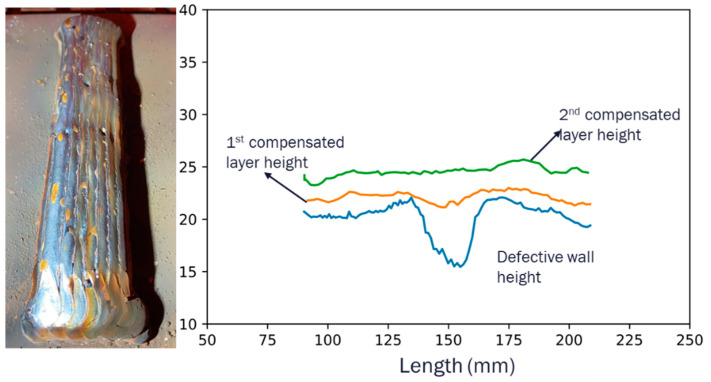
Compensated parallel wall deposited by the CW-MIG process.

**Table 1 sensors-23-08498-t001:** Process parameter range used in the experiment.

Parameters	Minimum Value	Maximum Value
Wire feed speed (WFS)	60 mm/s	90 mm/s
Current	200 Amp	250 Amp
Travel speed (TS)	6 mm/s	10 mm/s

**Table 2 sensors-23-08498-t002:** Control points.

Index	CP#0 *	CP#1	CP#2	CP#3	CP#4	CP#5	CP#6	CP#7	CP#8	CP#9 *
Distance (mm) *	0.0	50.0	60.0	140.0	160.0	235.0	265.0	310.0	350.0	400.0

* This distance is referred to as the wall start point; CP#0 is the start of the deposition wall, and CP#9 is the endpoint.

**Table 3 sensors-23-08498-t003:** Experimental plan regarding the process parameter setup.

Wall Index	Process Parameters for Base Layers			Process Parameters for CUs			Expected Wall Height
	WFS (mm/s)	Current (Amp)	TS (mm/s)	WFS (mm/s)	Current (Amp)	TS (mm/s)	
1	60	200	6	75	200	6	Increment
2				90			
3		250		75	250		
4				90			
5	90	200		60	200		Decrement
6				75			
7		250		60	250		
8				75			

**Table 4 sensors-23-08498-t004:** Experimental plan for the transition area study.

Wall Index	Process Parameters for Base Layers			Process Parameters for CUs		
	WFS (mm/s)	Current (Amp)	TS (mm/s)	WFS (mm/s)	Current (Amp)	TS (mm/s)
1	60	200	10	80	200	10
2	75	225		60	225	
3	60	250	8	80	250	8
4	75	250		60	250	

**Table 5 sensors-23-08498-t005:** Comparison of the control unit region within different travel speeds.

TS	CU: 10 mm	CU: 20 mm	CU: 30 mm	CU: 40 mm
6 mm/s	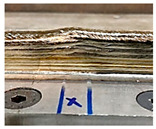	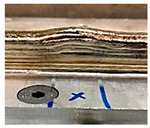	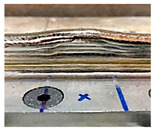	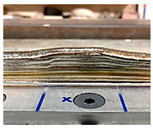
8 mm/s	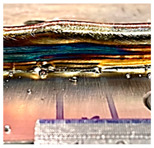	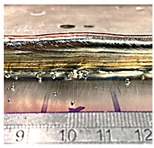	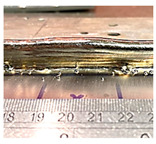	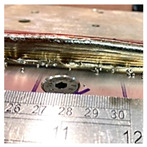
10 mm/s	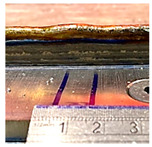	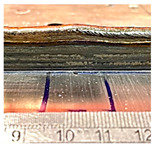	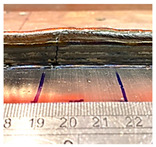	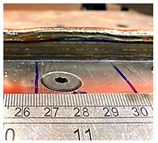

**Table 6 sensors-23-08498-t006:** Validation of initial wall compensation.

Layer No.	Wall Photos	Average Variation (mm)
13	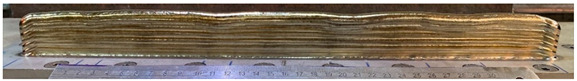	−0.64
14	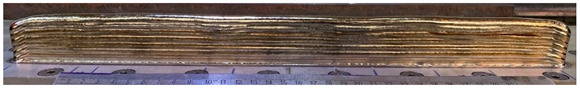	−0.19
15	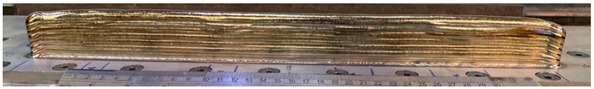	0.06
16	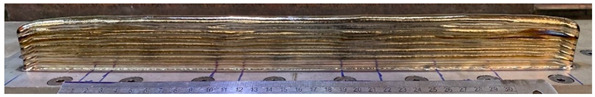	0.39

**Table 7 sensors-23-08498-t007:** CPs determined by the RRI sensor.

Index	Initial CP	CPs for the 14th Layer	Error (mm)	CPs for the 15th Layer	CPs for the 16th Layer	CPs for the 17th Layer
CP #0	0	0	-	0	0	0
CP #1	50	-	-	55.3		
CP #2	60	-	-	76.2	83.2	82.4
CP #3	140	139	−1 mm	138.3		
CP #4	160	165	+5 mm	174.9		
CP #5	235	236	+1 mm	232.9		
CP #6	265	268	+3 mm	276.8		
CP #7	310	311	+1 mm	308.1	362.8	
CP #8	350	351	+1 mm	368	380.3	354.2
CP #9	400	400	0	400	400	400

**Table 8 sensors-23-08498-t008:** Wall layer 13’s height variation with control.

Layer	Wall Photo	Average Variation (mm)
13	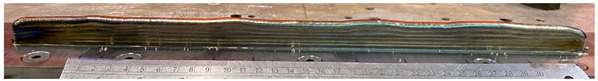	1.01
14	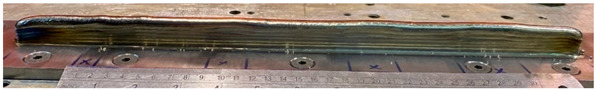	0.66
15	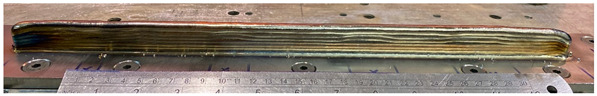	0.23
17	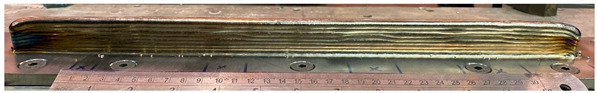	0.06

## Data Availability

Data underlying this study can be accessed through the Cranfield University repository at https://cord.cranfield.ac.uk/account/articles/24311296 (accessed on 13 September 2023).

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
