# Peer review of "Automated Interlayer Wall Height Compensation for Wire Based Directed Energy Deposition Additive Manufacturing"

_sensors, 2023, doi:10.3390/s23208498_

Round 1
Reviewer 1 Report
Review
The manuscript “Automated Interlayer Wall Height Compensation for Wire based Directed Energy Deposition Additive Manufacturing (w-DEDAM)” is devoted to the study of wall formation during the additive manufacturing of products from raw material wire in the presence of disturbances caused by fluctuations in the height of the deposited layer of various lengths. The work developed a system for monitoring the height of the beads and algorithms for automatic correction of depositing modes to stabilize the height of the wall. Taking into account the active development of additive manufacturing methods and their implementation in industrial production, the work is relevant and of practical interest.
The authors developed an experimental research stand that included a Coherent Range-Resolved Interferometry (Co-RRI) sensor, which showed the possibility of stable operation under conditions of disturbances caused by depositing wire with a plasma heat source. The system and control algorithms developed by the authors demonstrated the possibility of correcting depositing defects over several following layers.
Notes:
1. Figures 6 and 7 are identical, only the captions and indicated modes differ, apparently the authors made a mistake when preparing the figures.
2. The authors do not describe the methodology for determining the impact of transition time on parameter changes in paragraph 3.2.2, and therefore this point remains vague. In addition, the data collection frequency of 4 Hz may not be sufficient to identify process features.
3. It is possible to use higher-frequency acquisition systems, as well as joint analysis of current oscillograms, wire feed speed, travel speed and changes in bead height will help more accurately adjust the compensation system for different longitudinal sizes of the correction area.
The work is a completed study and can be published after minor modifications.
Author Response
Reviewer #1:
The manuscript “Automated Interlayer Wall Height Compensation for Wire based Directed Energy Deposition Additive Manufacturing (w-DEDAM)” is devoted to the study of wall formation during the additive manufacturing of products from raw material wire in the presence of disturbances caused by fluctuations in the height of the deposited layer of various lengths. The work developed a system for monitoring the height of the beads and algorithms for automatic correction of depositing modes to stabilize the height of the wall. Taking into account the active development of additive manufacturing methods and their implementation in industrial production, the work is relevant and of practical interest.
The authors developed an experimental research stand that included a Coherent Range-Resolved Interferometry (Co-RRI) sensor, which showed the possibility of stable operation under conditions of disturbances caused by depositing wire with a plasma heat source. The system and control algorithms developed by the authors demonstrated the possibility of correcting depositing defects over several following layers.
Notes:
- Figures 6 and 7 are identical, only the captions and indicated modes differ, apparently the authors made a mistake when preparing the figures.
Response: Thank you very much for the question. We do apologise our mistakes. We have revised the Figure 7 accordingly and review other figures to avoid the same mistakes. The changes have been highlighted in the revised manuscript.
- The authors do not describe the methodology for determining the impact of transition time on parameter changes in paragraph 3.2.2, and therefore this point remains vague. In addition, the data collection frequency of 4 Hz may not be sufficient to identify process features.
Response: Thanks for the comments. The methodology for determining the impact of transition time is similar to the previous experiment. Tale 5 shows the details of parameter changes for this purpose. We also insert some explanation regarding the methodology in Section 3.2.2 for the clarification.
‘The methodology for assessing the impact of process parameters on transition time follows the same approach as our previous experiments. Various parameters are applied to the control units (CUs), and the dimensions of the transition area are then measured. Based on the robot's travel speed, we are able to calculate the transition time.’
Thank you for highlighting data collection frequency. The data collection frequency of 4 Hz, mentioned in Section 3.2, is specifically used to evaluate the wire feed speed control response. This data is gathered through our system monitoring software, which operates at an acquisition rate of 4 Hz. While this is a relatively low frequency, it is sufficient for capturing changes in wire feed speed over intervals greater than 0.5 seconds. For other data, such as wall height, we used a much higher frequency of 3.2 kHz, as discussed in Section 2.2. We appreciate your attention to this detail, as it could potentially cause confusion. We have made slight revisions to this section to clarify this aspect.
- It is possible to use higher-frequency acquisition systems, as well as joint analysis of current oscillograms, wire feed speed, travel speed and changes in bead height will help more accurately adjust the compensation system for different longitudinal sizes of the correction area.
Response: Yes, you are right. The higher frequency acquisition does help the accuracy of the results in the transition time study. This have been mentioned in discussion section.
The work is a completed study and can be published after minor modifications.
Response: Thanks again for your valuable feedback and comments which are very valuable, also help us to highlight the future works regarding this topic.

Reviewer 2 Report
In this research paper titled “Automated layer-to-layer wall height compensation for wire-based directed energy deposition additive manufacturing (w-DEDAM),” the authors effectively address a significant challenge in additive manufacturing. This study demonstrates a commendable approach to monitoring and controlling changes in interlayer deposition height, particularly in transition regions. The utilization of coherent range-resolved interferometry (Co-RRI) sensor technology is a noteworthy contribution to the field. This research is promising, but further elaboration and contextualization will strengthen its impact and relevance in the additive manufacturing field. However, some formatting issues still need attention in the article. Therefore, it is recommended that the article be revised and accepted.
1. It is recommended not to put the English abbreviation of the article title in the title of the paper.
Author Response
Response: Thank you for your valuable comments and suggestions. We've updated the article title in line with your feedback. We greatly appreciate your endorsement.

Reviewer 3 Report
(1)As for FIG. 6 and FIG. 7, the paper does not discuss the law of wall height change caused by the change of different parameters. Can you discuss the law of wall height change when the parameters change?
(2) In Figure 10 (c), with the increase of WFS/TS, the transition time increases, but when WFS/TS is equal to 2.0, the transition time decreases to 1.813, which is not in line with the overall upward trend. What is the reason for this data?
(3) Can you provide more details on the implementation of the Co-RRI sensor? How was the sensor calibrated and how were the measurements obtained?
(4) Have you conducted any comparisons with other existing methods for part quality monitoring and control in additive manufacturing? How does the proposed method compare in terms of accuracy and efficiency?
Author Response
Reviewer #3:
(1)As for FIG. 6 and FIG. 7, the paper does not discuss the law of wall height change caused by the change of different parameters. Can you discuss the law of wall height change when the parameters change?
Response: Thank you for your insightful comments. We realised that Figure 7 was incorrectly attached and have since rectified it. Regarding the impact of process parameters on part geometry, specifically wall height and wall width, we didn't delve too deeply into this aspect. This is because the process work has already been comprehensively studied and documented in Dr Martina's article [1]. The process parameters we employed in our manuscript are in line with this research, as highlighted in Section 3.1. However, we do value your suggestion and have added some relevant explanations in Section 3.2.1 to address this.
‘In general, when both the current and travel speed are held constant, an increase in wire feed speed (WFS) leads to a corresponding rise in wall height, while a decrease in WFS results in a lower wall height. It's worth noting that, based on the work of Martina et al [1], the wall width remains largely consistent when using the process parameters under consideration.’
(2) In Figure 10 (c), with the increase of WFS/TS, the transition time increases, but when WFS/TS is equal to 2.0, the transition time decreases to 1.813, which is not in line with the overall upward trend. What is the reason for this data?
Response: Thank you for bringing this to our attention. We also observed this anomaly during our experiments. We attribute it to various uncertainties that can occur during experimental procedures, such as errors in measurement, recording, and device inconsistencies. We've addressed this outlier in Section 3.2.2 of our paper, where we provide a detailed explanation. Additionally, we've included some discussion on future work that aims to explore these uncertainties further.
‘When the ratio of wire feed speed (WFS) to travel speed (TS) increases, either due to higher WFS or lower TS, the transition time also correspondingly increases. As observed in Figure 10(c), when the WFS to TS ratio is 2.0, the recorded transition time deviates from the trend line established by the other four recorded results. This anomaly can be attributed to experimental uncertainties, such as measurement errors, time recording discrepancies, and inconsistencies in monitoring devices. Further discussion on these experimental uncertainties can be found in Section 5.’
(3) Can you provide more details on the implementation of the Co-RRI sensor? How was the sensor calibrated and how were the measurements obtained?
Response: Thanks for the recommendation. We have inserted/revised the relevant contents in Section 2.2.
‘In this technology, shown as Figure 3, a diode laser's optical frequency is sinusoidally modulated, and the resulting light is both supplied to and collected from the target using optical fibre. The light reflected off the target (usually the layer surface) interferes with reference light reflected from the fibre tip. The resulting light is demodulated for a range of fibre-tip to target-surface distances covering approximately 100mm using pre-computed time-variant carrier signals that correspond to the expected interference signals. The absolute distance of the target is then determined by evaluating the return signal intensity as a function of range using a Gaussian peak fit. The complete mathematical derivation has been presented in Ref [18].
The proposed RRI sensor offers a robust and versatile method for distance measurements. Implemented in a reduced-sized configuration for a more compact assembly head it is capable of achieving a measurement resolution of better than 100 μm and a working range exceeding 100 mm. As an added benefit, given its coherent nature, it is inherently insensitive to arc light, allowing for measurements close to the weld pool [19]. Furthermore, to obtain the accurate measurement, the RRI was also calibrated using a Calibration Block - a pre-designed precision block with stairs of varying known dimensions. The Collimator is placed at a working distance from the surface of each stair landing and the range measurement is taken. The ratio of the measured range against the known stair dimension is used to calibrate the sensor.’
(4) Have you conducted any comparisons with other existing methods for part quality monitoring and control in additive manufacturing? How does the proposed method compare in terms of accuracy and efficiency?
Response: That's an excellent question, and I appreciate the opportunity to clarify. For the work on wall height compensation, we didn't make comparisons with other methods, primarily because we haven't come across any similar published work specifically focused on wire arc-based directed energy deposition processes, especially focusing on the inter layer compensation. As for the accuracy of the RRI sensor, this is detailed in Ref [2] and further elaborated upon in Section 2.2 of our paper. In the discussion section, we also applied our proposed methodology to another wire arc directed energy deposition process, and it performed exceptionally well. Moreover, this method is currently under industrial development for application in real-world part building scenarios. The proposed control flowchart is designed to be automatically integrated into the process. We are confident that our method meets the accuracy and efficiency requirements of the industry.
Reference:
- Martina, F., Mehnen, J., Williams, S.W., Colegrove, P., Wang, F.: Investigation of the benefits of plasma deposition for the additive layer manufacture of Ti–6Al–4V. J Mater Process Technol. 212, 1377–1386 (2012). https://doi.org/10.1016/J.JMATPROTEC.2012.02.002
- Hallam, J.M., Kissinger, T., Charrett, T.O.H., Tatam, R.P.: In-process range-resolved interferometric (RRI) 3D layer height measurements for wire + arc additive manufacturing (WAAM). Meas Sci Technol. 33, (2022). https://doi.org/10.1088/1361-6501/ac440e

Round 2
Reviewer 3 Report
It can be accepted.